# Learning with Interaction: Agentic Distillation for Large Language Model Reasoning

## Abstract

Recent advancements in large language models (LLMs) have demonstrated remarkable reasoning abilities to solve complex tasks. However, these gains come with significant computational costs, limiting their practical deployment. A promising direction is to distill reasoning skills from larger teacher models into smaller, more efficient student models, yet existing data-centric distillation approaches suffer from passive learning, over-learning on simple tasks, and persistent knowledge gaps. To overcome these limitations, we introduce AGENTIC DISTILLATION, a novel framework for adaptive and active distillation. In AGENTIC DISTILLATION, student LLMs interact with teacher LLMs modeled as environments, receiving feedback tokens to guide their reasoning process and selectively updating their capabilities when necessary. To address the off-policy and gradient vanishing challenges introduced by feedback tokens, we devise a tailored importance sampling and clipping strategy within a unified objective that both incentivizes reasoning and injects knowledge into student LLMs. Extensive experiments show that AGENTIC DISTILLATION significantly enhances distillation performance, offering a scalable path for equipping compact LLMs with advanced reasoning abilities.

## 1 Introduction

In recent years, large language models (LLMs) have undergone rapid advancements, showcasing exceptional performance across various natural language processing tasks (Pu et al., 2023; Zhang & Soh, 2024; Gupta et al., 2024; Xu et al., 2024). In particular, LLMs employing long chain-of-thought (CoT) reasoning have demonstrated remarkable proficiency in solving intricate problems spanning mathematics, coding, and science, significantly pushing the boundaries of complex capabilities in foundation models (OpenAI, 2024a;b; 2025; DeepSeek-AI et al., 2025; Kimi-Team et al., 2025; Yang et al., 2025; Comanici et al., 2025; Huang & Yang, 2025; xAI, 2025).

However, the enhanced model reasoning capability introduces increased computational costs. The growth in model parameters and the extended length of CoT reasoning elevate computational demands, limiting practical applications (Chen et al., 2025). Therefore, *equipping more efficient small language models with robust reasoning capabilities via learning from stronger large models* has garnered significant attention from researchers and the broader community (DeepSeek-AI et al., 2025; Wen et al., 2025; Muennighoff et al., 2025; Guha et al., 2025; Ye et al., 2025).

A prevalent approach is *data-centric distillation*, which employs *rejection sampling* (Wang et al., 2024; Yang et al., 2024b; Shao et al., 2024; Ying et al., 2024) to generate training trajectory-level data for distilling student models. This method generates multiple reasoning trajectories from stronger LLMs for a given query, selects those with correct conclusions, and uses them to train the student

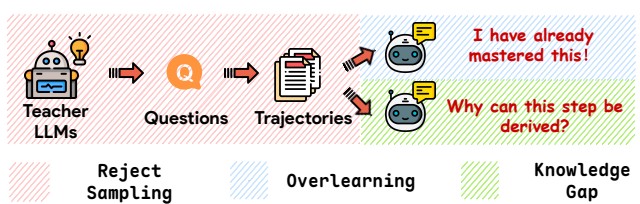

Figure 1: Illustration of *over-learning* and *knowledge gap* issues in data-centric distillation.

model via supervised finetuning (SFT). (Qin et al., 2024; DeepSeek-AI et al., 2025) or reinforcement learning (RL) (Zhang et al., 2025a). This enables student models to acquire the knowledge and

reasoning capabilities of teacher LLMs. However, this approach has notable limitations due to its characteristic of **passive learning** as shown in Figure 1: 1) **Over-Learning**: Training on static and complete trajectories from teacher LLMs does not dynamically adapt to the evolving capabilities of the student model, often leading to over-learning on simpler questions (Chu et al., 2025) and wasting data and training resources on mastered questions.; 2) **Knowledge Gap**: Teacher-centered data generation approaches may neglect the specific knowledge requirements and competency gaps of student LLMs (Liu et al., 2024a). Much like a standardized teaching approach that overlooks individual student deficiencies, reasoning processes that appear intuitive to advanced teacher models can be inaccessible to student models. As a result, student models tend to merely mimic the stylistic features of the teacher's output rather than acquiring robust reasoning capabilities (Chu et al., 2025; Kirk et al., 2024; Wu et al., 2025).

To address these limitations, we propose AGENTIC DISTILLATION, a novel framework for distilling knowledge and reasoning capabilities from strong LLMs into smaller student LLMs through **active** and **adaptive interaction**. Unlike traditional distillation methods that passively transfer knowledge, AGENTIC DISTILLATION empowers the student LLM to dynamically determine when to query the teacher LLM during reasoning, seeking feedback only when necessary as shown in Figure 2. This enables the student LLM to refine its reasoning process based on teacher feedback, leading to more accurate outcomes. Additionally, we design a mechanism to allow the student LLM to effectively learn essential knowledge and reasoning abilities from the teacher LLM's feedback.

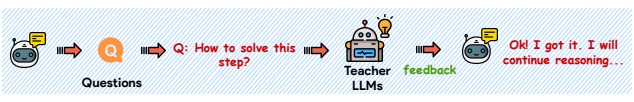

Figure 2: Simple Illustration of AGENTIC DISTILLATION.

To tackle off-policy and gradient vanishing issues inherent in learning from feedback tokens, we introduce a tailored importance sampling coefficient and clipping strategy. They are seamlessly integrated into a unified objective that both incentivizes reasoning and injects knowledge into student LLMs. Notably, recent works (e.g., Search-R1) (Wang et al., 2025; Singh et al., 2025; Jin et al., 2025; Liu et al., 2025) primarily focus on enhancing LLMs' interactions with external environments (e.g., tools), often overlooking the rich information embedded in feedback, which can be utilized to improve the reasoning ability of student LLMs themselves. In contrast, AGENTIC DISTILLATION leverages teacher feedback as a direct learning signal, enabling continuous improvement of the student model. Even without interaction during inference, AGENTIC DISTILLATION-trained student LLMs can successfully reason on previously unsolvable tasks.

We conduct extensive experiments to validate the effectiveness of AGENTIC DISTILLATION. For example, AGENTIC DISTILLATION enhances the performance of Qwen2.5-7B-Instruct on mathematical reasoning benchmarks, achieving an average improvement of approximately 4 points over baseline distillation strategies. Significant gains are also observed on out-of-domain benchmarks, demonstrating AGENTIC DISTILLATION's robust generalization. Additional experiments confirm that AGENTIC DISTILLATION generalizes effectively across various student and teacher LLMs. Additionally, we investigate whether AGENTIC DISTILLATION expands the knowledge boundaries of student LLMs. Analysis of training dynamics and student LLM responses reveals that AGENTIC DISTILLATION enables student LLMs to effectively acquire new knowledge and capabilities, aligning their reasoning abilities with those of teacher LLMs.

## 2 METHOD

In this section, we introduce the motivation to propose the AGENTIC DISTILLATION framework for adaptive and active distillation (§ 2.1). Then, we introduce the details of proposed AGENTIC DISTILLATION (§§ 2.2 and 2.3)

### 2.1 PRELIMINARIES

**Distillation from Strong LLMs.** In a typical LLM reasoning task, given a question $q$ from the question distribution $q \sim P(Q)$, the LLM $\pi_\theta$ is prompted with an instruction $I$ to generate an answer:

$$a' \leftarrow \pi_\theta(\cdot \mid q, I). \tag{1}$$

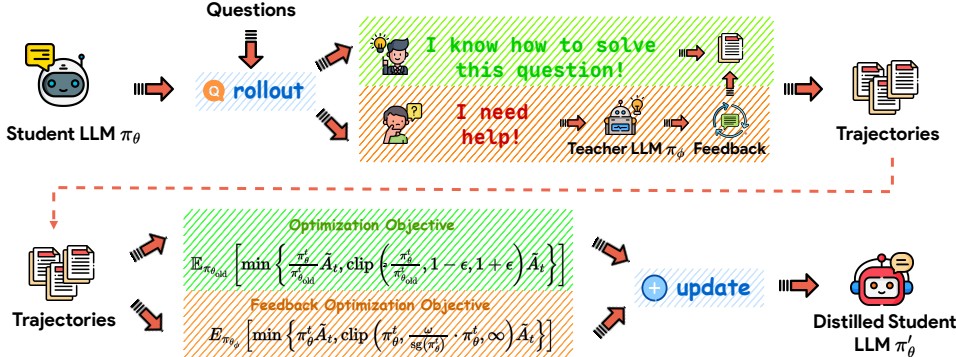

Figure 3: Illustration of AGENTIC DISTILLATION. In the AGENTIC DISTILLATION framework, during each rollout process, the student LLM initially attempts to solve a given question independently. If the student LLM fails to resolve the question, it engages in *external interaction* by querying the teacher LLM for feedback. Otherwise, the student LLM proceeds with its reasoning to derive the final answer. Subsequently, we compute the reward and optimize the student LLM using losses derived from both internal tokens and feedback tokens, respectively.

For chain-of-thought based LLM reasoning, the reasoning process involves a step-by-step sequence, typically enclosed within tags such as `<think>` and `</think>`, represented by the token sequence $\tau_{\text{thinking}}$. This culminates in a final conclusion $\tau_{\text{conclusion}}$, which includes the predicted answer $a'$:

$$[\tau_{\text{thinking}}, \tau_{\text{conclusion}}] \leftarrow \pi_\theta(\cdot \mid q, I). \tag{2}$$

The objective of this paper is to distill knowledge and capabilities from a strong teacher LLM $\pi_\phi^t$ to enhance a student LLM $\pi_\theta^s$:

$$\pi_\theta^{s'} \leftarrow \text{D}(\pi_\theta^s, \pi_\phi^t, q), \tag{3}$$

where D represents the distillation method, such as passive data-centric distillation (Qin et al., 2024; DeepSeek-AI et al., 2025; Wen et al., 2025) or the AGENTIC DISTILLATION proposed in this paper.

**Data-Centric Distillation.** The predominant distillation approach is *rejection sampling* (Yang et al., 2024b; Shao et al., 2024; Guha et al., 2025; Wen et al., 2025). Specifically, given a question set $\mathcal{Q}$, a strong teacher LLM $\pi_t$ generates predictions $\mathcal{T}$ for each question $q \in \mathcal{Q}$:

$$\tau \sim \pi_\phi^t(\cdot \mid q, I), \quad \tau \in \mathcal{T}, q \in \mathcal{Q}. \tag{4}$$

The prediction set $\mathcal{T}$ is then filtered based on the correctness of each prediction:

$$\mathcal{T}' = \left\{ \tau \mid \mathbb{I}(a, a') \right\}, \tag{5}$$

where $a$ denotes the ground truth answer to $q$ and $\mathbb{I}$ is an indicator function that returns 1 only when the prediction is correct. The selected predictions $\mathcal{T}'$ are used to train the student LLM $\pi_s$:

$$\mathcal{L}(\theta) = \mathbb{E}_{q \in \mathcal{Q}, \tau \sim \mathcal{T}'} \left[ -\log \pi_\theta^s(\tau \mid q; \theta) \right], \tag{6}$$

where $\theta$ denotes the parameters of the student LLM $\pi_s$.

**Distillation from Interaction.** To address the limitations of passive data-centric distillation as mentioned in § 1, we propose a novel approach that distills knowledge through active interaction with the teacher LLM. Specifically, we augment the reasoning process $\tau_{\text{thinking}}$ to include multiple turns of interaction, comprising *queries to the teacher LLM $\tau_q$* and *external feedback from the teacher LLM $\tau_o$*, formally expressed as:

$$\left[ \ldots, \tau_{q,(1)}, \tau_{o,(1)}, \ldots, \tau_{q,(N)}, \tau_{o,(N)}, \ldots \right] \leftarrow \tau_{\text{thinking}}. \tag{7}$$

The mechanisms governing interaction with the teacher LLM and the process of learning from its feedback are detailed in § 2.2 and § 2.3, respectively.

## 2.2 AGENTIC INTERACTION WITH TEACHER LLMs.

To distill knowledge from the teacher LLM, we design an agentic interaction mechanism that enables the student LLM to actively and flexibly interact with the teacher during the reasoning process.

When faced with a question $q$, the student LLM first performs basic reasoning using its internal knowledge, such as problem decomposition, solution planning, and simple arithmetic operations (Wei et al., 2022). If the student LLM can solve the question using only its own knowledge and reasoning abilities, we argue that external knowledge distillation from an *oracle* is unnecessary. This important distinction is often overlooked by typical SFT-based methods (Qin et al., 2024; Huang et al., 2024; Muennighoff et al., 2025; Guha et al., 2025). Conversely, during reasoning, when the student recognizes that a (sub-)question exceeds the limits of its internal knowledge, it must refer to external oracle information. In such cases, we allow the student LLM to query the teacher LLM in natural language.

Specifically, we provide the student LLM with the prompt shown in Prompt 2.1 (full version is provided in Prompt A.1), which instructs it to enclose natural language queries to the teacher within `<query>` and `</query>` tags. The teacher LLM then responds with the corresponding answer (i.e., *feedback* or *observation*), appended to the student's reasoning process within `<result>` and `</result>` tags. To avoid meaningless or inefficient loops, we also impose an interaction budget limiting the number of queries the student may direct to the teacher.

---

**Prompt 2.1: Prompt to Equip Student LLM with Agentic Interaction Capability**

**Reasoning Process**

- **Decomposition:** Break down the user's question into a logical, step-by-step sequence of reasoning. Start from the most basic facts and build upon them.
- **External Inquiry (Optional but Encouraged):**
  - You may issue up to *max_turns* queries to an External Environment to validate hypotheses, clarify information, or advance your reasoning.
  - Each query must be a self-contained question enclosed in `<query>...</query>` tags.
  - **Wait for the `<result>...</result>` block** from the environment before continuing your reasoning.
  - **Critically analyze and integrate** the content from the `<result>...</result>` block into your reasoning chain.
  - Do not invent, assume, or hallucinate any `<result>` content. Your reasoning must be grounded in the provided results.

---

## 2.3 LEARNING FROM AGENTIC INTERACTION

This section addresses learning from agentic interaction. Given a query $\tau_v$ generated by the student LLM and feedback $\tau_o$ provided by the teacher LLM, prior RL approaches typically exclude $\tau_o$ from the loss calculation (Song et al., 2025; Liu et al., 2025), as the student LLM is not expected to generate tokens from the external environment. In contrast, our approach integrates feedback tokens $\tau_{o,(1:T_o)}$ into the RL policy loss to enable the student LLM to acquire new knowledge and capabilities.

The classical clipped surrogate objective is defined as:

$$
\mathcal{J}(\theta) = \mathbb{E}_{q \sim P, \{\tau_i\} \sim \pi_\theta} \left[ \frac{1}{G} \sum_{i=1}^{G} \frac{1}{|\tau_i|} \sum_{t=1}^{|\tau_i|} \left\{ \min \left( \rho_{i,t} \tilde{A}_t^i, \text{clip}(\rho_{i,t}, 1 - \epsilon, 1 + \epsilon) \tilde{A}_t^i \right) \right. \right.
$$
$$
\left. \left. - \beta D_{\text{KL}} \big[ \pi_\theta \| \pi_{\text{ref}} \big] \right\} \right], \tag{8}
$$

where the importance sampling coefficient for each token $\tau_{i,(t)}$ at index $t$ is given by:

$$
\rho_{i,t} = \frac{\pi_\theta \left( \tau_{i,(t)} \mid \tau_{i,(\leq t)} \right)}{\pi_{\theta_{\text{old}}} \left( \tau_{i,(t)} \mid \tau_{i,(\leq t)} \right)}, \tag{9}
$$

and $\pi_{\theta_{\text{old}}}$ denotes the previous policy of the student LLM. Directly applying this loss to feedback tokens $\tau_{o,(1:T_o)}$ may introduce the off-policy error due to the mismatch between feedback tokens and

the student LLM's policy, which can destabilize RL training (Schulman et al., 2017; Zhang et al., 2025a).

**Amending Importance Sampling Coefficient for Feedback Tokens.** To mitigate the off-policy error, we introduce a modified importance sampling coefficient $\tilde{\rho}$. Within the standard clipped surrogate loss, the off-policy error stems from sampling the trajectory $\tau$ from $\pi_{\theta_{\text{old}}}$:

$$\mathcal{J}(\theta) = \mathbb{E}_{\boldsymbol{\tau} \sim \pi_{\theta_{\text{old}}}} \left[ \frac{1}{|\boldsymbol{\tau}|} \sum_{t=1}^{|\boldsymbol{\tau}|} \frac{\pi_\theta\big(\boldsymbol{\tau}_{(t)}|\boldsymbol{\tau}_{(\leq t)}\big)}{\pi_{\theta_{\text{old}}}\big(\boldsymbol{\tau}_{(t)}|\boldsymbol{\tau}_{(\leq t)}\big)} \tilde{A}_t \right], \tag{10}$$

where clipped and KL-penalty terms are omitted for simplicity. However, feedback tokens follow the distribution $\pi_\phi$, defined by the teacher LLM, leading to the modified objective:

$$\mathcal{J}(\theta) = \mathbb{E}_{\pi_{\theta_{\text{old}}}} \left[ \frac{1}{N} \sum_{\boldsymbol{\tau}_{(t)} \notin \boldsymbol{\tau}_o} \frac{\pi_\theta\big(\boldsymbol{\tau}_{(t)}|\boldsymbol{\tau}_{(\leq t)}\big)}{\pi_{\theta_{\text{old}}}\big(\boldsymbol{\tau}_{(t)}|\boldsymbol{\tau}_{(\leq t)}\big)} \tilde{A}_t \right] + \mathbb{E}_{\pi_\phi} \left[ \frac{1}{M} \sum_{\boldsymbol{\tau}_{(t)} \in \boldsymbol{\tau}_o} \frac{\pi_\theta\big(\boldsymbol{\tau}_{(t)}|\boldsymbol{\tau}_{(\leq t)}\big)}{\pi_\phi\big(\boldsymbol{\tau}_{(t)}|\boldsymbol{\tau}_{(\leq t)}\big)} \tilde{A}_t \right], \tag{11}$$

where $N$ and $M$ are the number of non-feedback tokens and feedback tokens, respectively. Directly using the teacher LLM's distribution to compute the importance sampling coefficient is a straightforward approach but has two limitations: 1) vocabulary differences between the teacher and student LLMs may cause inconsistent distributions, and 2) computing the teacher LLM's distribution incurs additional computational overhead. To address these, we propose treating the teacher LLM's distribution as a one-hot distribution, yielding:

$$\mathcal{J}(\theta) = \mathbb{E}_{\pi_{\theta_{\text{old}}}} \left[ \frac{1}{N} \sum_{\boldsymbol{\tau}_{(t)} \notin \boldsymbol{\tau}_o} \frac{\pi_\theta\big(\boldsymbol{\tau}_{(t)}|\boldsymbol{\tau}_{(\leq t)}\big)}{\pi_{\theta_{\text{old}}}\big(\boldsymbol{\tau}_{(t)}|\boldsymbol{\tau}_{(\leq t)}\big)} \tilde{A}_t \right] + \mathbb{E}_{\pi_\phi} \left[ \frac{1}{M} \sum_{\boldsymbol{\tau}_{(t)} \in \boldsymbol{\tau}_o} \pi_\theta\big(\boldsymbol{\tau}_{(t)}|\boldsymbol{\tau}_{(\leq t)}\big) \tilde{A}_t \right]. \tag{12}$$

This method employs a temperature coefficient to sharpen the teacher LLM's distribution, reducing computational complexity and resolving vocabulary inconsistencies.

**Gradient Vanishing for Feedback Tokens.** The standard surrogate objective employs a clipping mechanism on the importance sampling coefficient to prevent excessive policy deviation from the previous policy. However, for feedback tokens, the importance sampling coefficient $\pi_\theta$ is inherently bounded due to the softmax activation. Consequently, we remove the standard clipping mechanism for feedback tokens. The gradient of these tokens can be computed as

$$\pi_\theta \cdot \tilde{A}_t \cdot \nabla_\theta \cdot \log \pi_\theta. \tag{13}$$

Nevertheless, when the probability of a feedback token $\tau_{o,(t)}$ in the student LLM's policy is low ($\pi_\theta(\tau_{o,(t)}) \to 0$), the gradient approaches zero ($\pi_\theta \cdot \tilde{A}_t \cdot \nabla_\theta \log \pi_\theta \to 0$). This vanishing gradient leads to suboptimal learning, particularly for off-policy feedback tokens from the teacher LLM, which are critical for the student LLM to learn effectively. These tokens often have low probabilities in the student LLM's policy, exacerbating the vanishing gradient issue and hindering knowledge transfer.

**Clipping Strategy for Feedback Tokens.** To address the vanishing gradient problem, we propose a clipping strategy inspired by the standard mechanism (Schulman et al., 2017):

$$\text{clip}\left(\pi_\theta, \frac{\omega}{\text{sg}\,(\pi_\theta)} \cdot \pi_\theta, \infty\right), \tag{14}$$

where $\omega$ is a clipping hyperparameter and $\text{sg}(\cdot)$ denotes the stop-gradient operation. This approach sets a lower bound on the importance sampling coefficient for feedback tokens, with the $\pi_\theta/\text{sg}(\pi_\theta)$ term ensuring numerical equivalence. The resulting gradients are:

$$\begin{cases} \pi_\theta \cdot \tilde{A}_t \cdot \nabla_\theta \cdot \log \pi_\theta, & \text{if } \pi_\theta \geq \omega, \\ \omega \cdot \tilde{A}_t \cdot \nabla_\theta \cdot \log \pi_\theta, & \text{if } 0 \leq \pi_\theta < \omega. \end{cases} \tag{15}$$

This ensures that feedback tokens with high advantage maintain non-vanishing gradients, mitigating the impact of policy deviation.

**Final Objective.** By integrating the modified importance sampling coefficient and the proposed clipping strategy for mitigating the off-policy and the vanishing gradient issues, we formulate the final objective for optimizing the student LLM:

$$\mathcal{J}(\theta) = \mathbb{E}_{\pi_{\theta_{\text{old}}}} \left[ \min \left\{ \frac{\pi_\theta^t}{\pi_{\theta_{\text{old}}}^t} \tilde{A}_t, \text{clip} \left( \frac{\pi_\theta^t}{\pi_{\theta_{\text{old}}}^t}, 1 - \epsilon, 1 + \epsilon \right) \tilde{A}_t \right\} \right]$$
$$+ \mathbb{E}_{\pi_\phi} \left[ \min \left\{ \pi_\theta^t \tilde{A}_t, \text{clip} \left( \pi_\theta^t, \frac{\omega}{\text{sg}\left(\pi_\theta^t\right)} \cdot \pi_\theta^t, \infty \right) \tilde{A}_t \right\} \right], \quad (16)$$

where $\pi_\theta^t = \pi_\theta(\tau_{(t)}|\tau_{(\leq t)})$. Intuitively, the objective of AGENTIC DISTILLATION unifies the RLVR and SFT in a single function. For action tokens autonomously generated by student LLMs, such as problem decomposition, solving, and query formulation, we employ the standard RLVR objective function for optimization (*first part*). For feedback tokens provided by teacher LLMs, we adopt an SFT-inspired optimization objective, enhanced by advantage and clipping-controlled update (*second part*), to effectively inject new knowledge into the student LLMs.

## 3 EXPERIMENTS

### 3.1 SETUP

**Baselines.** We compare AGENTIC DISTILLATION against several representative LLM post-training methods: ❶ *Supervised Fine-Tuning (SFT)*: Utilizes teacher LLM-generated data through rejection sampling; ❷ *Vanilla Reinforcement Learning (RL)*: Trains the student LLM using the GRPO algorithm (Shao et al., 2024) without external environment interactions; ❸ *Reinforcement Learning with Supervised Fine-Tuning (RL+SFT)*: Combines GRPO training (Shao et al., 2024) with data generated via rejection sampling; ❹ *Reinforcement Learning with Masked Interaction (RL+MI)*: Employs the GRPO algorithm (Shao et al., 2024) with teacher LLM interactions, but excludes feedback tokens from loss computation.

**Evaluation Benchmarks.** We evaluated all models across four domain-specific benchmarks: ❶ *Mathematical Reasoning*: Includes AIME24, AIME25, MATH500 (Hendrycks et al., 2021), and LiveMathBench (Liu et al., 2024b); ❷ *Scientific Reasoning*: Represented by GPQA-Diamond (Rein et al., 2023); ❸ *Code Reasoning*: Comprises MBPP (Austin et al., 2021) and LiveCodeBench (Jain et al., 2025); ❹ *Puzzle Reasoning*: Includes puzzles from Reasoning-Gym (Stojanovski et al., 2025).

**Implementation Details.** We conducted experiments on the Qwen-2.5 series models (Yang et al., 2024a) and Llama-3.2 series models (Dubey et al., 2024), distilling from two prominent teacher LLMs from distinct families: Qwen3-30B-A3B-Instruct-2507 (Yang et al., 2025) and GPT-OSS-20B (Agarwal et al., 2025). The training corpus, sourced from DAPO (Yu et al., 2025), OpenScienceReasoning-2 [1], and Reasoning Gym (Stojanovski et al., 2025), consists of approximately 60,000 high-quality reasoning-intensive samples. Models were trained for 200 steps with a batch size of 256 and a group size of 8, selecting the best model based on validation performance. During each generation, the student LLM was allowed up to three interactions with the teacher LLM. Training was performed using the veRL (Sheng et al., 2025) and vLLM (Kwon et al., 2023) frameworks. For evaluation, we set the sampling temperature to 1.0, top-$p$ to 1.0, top-$k$ to -1, and the maximum generation tokens to 16384. To reduce variance, we report average performance relative to the size of each benchmark. And the prompt utilized in inference phase is shown in Prompt A.2.

### 3.2 MAIN RESULTS AND ANALYSIS

Table 1 illustrates the performance of AGENTIC DISTILLATION and baselines on different benchmarks, containing different student LLMs and teacher LLMs. From the experimental results, we have the following findings.

**AGENTIC DISTILLATION Outperforms Baseline Methods.** As illustrated in Table 1, AGENTIC DISTILLATION surpasses other training strategies, including supervised fine-tuning (SFT), vanilla

---
[1] https://huggingface.co/datasets/nvidia/OpenScienceReasoning-2

Table 1: Experimental results of AGENTIC DISTILLATION and baselines with Qwen2.5-7B-Instruct as the student LLM. We report the average performance for 16 runs on AIME24 and AIME25, and 4 runs on others. We abbreviate LMB as LiveMathBench v202505, LCB as LiveCodeBench v6, RG as Reasoning Gym, MI as Masked Interaction, and AD as AGENTIC DISTILLATION. ♠ denotes the in-domain evaluation benchmark and ♣ denotes the out-of-domain benchmark. We provide performance of teacher LLMs in Table 4

| Methods | Math ♠ | | | | Science ♠ | Code ♣ | | Puzzle ♠ |
|---|---|---|---|---|---|---|---|---|
| | AIME24 Avg@16 | AIME25 Avg@16 | MATH500 Avg@4 | LMB Avg@4 | GPQA-D Avg@4 | MBPP Avg@4 | LCB Avg@4 | RG Avg@4 |
| Student LLM: *Qwen2.5-7B-Instruct*, Teacher LLM: *Qwen3-30B-A3B-Instruct-2507* | | | | | | | | |
| Original | 9.79 | 7.50 | 73.00 | 10.75 | 33.33 | 58.66 | 15.71 | 9.63 |
| +SFT | 11.67 | 13.54 | 75.80 | 12.00 | 22.35 | 41.34 | 11.04 | 18.98 |
| +RL | 12.08 | 10.00 | 75.35 | 10.50 | 34.72 | 58.27 | 16.33 | 19.81 |
| +RL+SFT | 13.01 | 12.13 | 75.77 | 11.32 | 35.20 | 59.22 | 15.34 | 19.22 |
| +RL+MI | 11.25 | 6.46 | 72.80 | 10.25 | 34.47 | 58.17 | 14.55 | 9.17 |
| +AD | **14.82** | **14.33** | **78.17** | **14.27** | **37.13** | **62.73** | **18.62** | **21.11** |
| Student LLM: *Qwen2.5-7B-Instruct*, Teacher LLM: *GPT-OSS-20B* | | | | | | | | |
| +SFT | 14.67 | 13.54 | 75.80 | 12.00 | 37.34 | 61.04 | **22.35** | 18.98 |
| +RL | 12.08 | 10.00 | 75.35 | 10.50 | 34.72 | 58.27 | 16.33 | 19.81 |
| +RL+SFT | 13.01 | 12.13 | 75.77 | 11.32 | 35.20 | 59.22 | 15.34 | 19.22 |
| +RL+MI | 12.31 | 9.26 | 71.89 | 11.33 | 31.25 | 56.25 | 13.79 | 10.25 |
| +AD | **16.52** | **17.47** | **81.22** | **16.29** | **38.53** | **64.15** | 20.27 | **24.32** |
| Student LLM: *Llama-3.2-3B-Instruct*, Teacher LLM: *Qwen3-30B-A3B-Instruct-2507* | | | | | | | | |
| Original | 2.50 | 1.20 | 30.10 | 3.00 | 23.48 | 42.61 | 6.87 | 9.44 |
| +SFT | 5.50 | 2.67 | **48.65** | 5.50 | 22.98 | 43.39 | 3.95 | 13.89 |
| +RL | 8.96 | 1.12 | 44.10 | 7.00 | 25.76 | 52.59 | 10.92 | 13.70 |
| +RL+SFT | 6.24 | 3.62 | 41.25 | 8.11 | 26.45 | 53.45 | 10.23 | 12.44 |
| +RL+MI | 7.44 | 2.98 | 45.11 | 6.45 | 24.26 | 54.27 | 9.84 | 10.52 |
| +AD | **10.38** | **4.42** | 44.45 | **9.00** | **28.54** | **58.66** | **15.70** | **16.85** |

reinforcement learning (RL), RL combined with SFT (RL+SFT), and RL with masked interaction (RL+MI). Notably, AGENTIC DISTILLATION achieves significant improvements on challenging reasoning benchmarks such as AIME24 and AIME25, with average accuracy gains of 4-6 points over the strongest baseline. Comparable enhancements are observed across science, code, and puzzle tasks, underscoring AGENTIC DISTILLATION's robustness in improving reasoning capabilities across diverse task settings.

**AGENTIC DISTILLATION Enhances Performance Across Diverse Student LLMs.** As depicted in Table 1, AGENTIC DISTILLATION consistently outperforms baseline methods across various student LLMs, including Qwen-2.5-7B-Instruct and Llama-3.2-3B-Instruct. The framework achieves stable improvements across LLMs of different architectures and sizes, highlighting the generality of AGENTIC DISTILLATION and its potential for broad application to diverse LLM families and types.

**AGENTIC DISTILLATION Improves Across Different Teacher LLMs.** AGENTIC DISTILLATION consistently delivers performance improvements across different teacher models, including the short-cot based reasoning LLMs *Qwen3-30B-A3B-Instruct-2507* and the long-cot based reasoning LLMs *GPT-0SS-20B*. While baseline methods exhibit variability depending on the teacher LLM, AGENTIC DISTILLATION maintains superior results, indicating that its adaptive training mechanism is independent of the teacher model. This stability highlights AGENTIC DISTILLATION's flexibility, making it suitable for scenarios with varying teacher quality or availability.

**AGENTIC DISTILLATION Generalizes to Out-of-Domain Benchmarks.** AGENTIC DISTILLATION also performs generalization across in-domain and out-of-domain benchmarks. On mathematics, science, and puzzle benchmarks, which align closely with the training data, AGENTIC DISTILLATION consistently outperforms all baselines. More notably, on out-of-domain benchmarks such as code (MBPP and LiveCodeBench), AGENTIC DISTILLATION achieves substantial gains, surpassing the

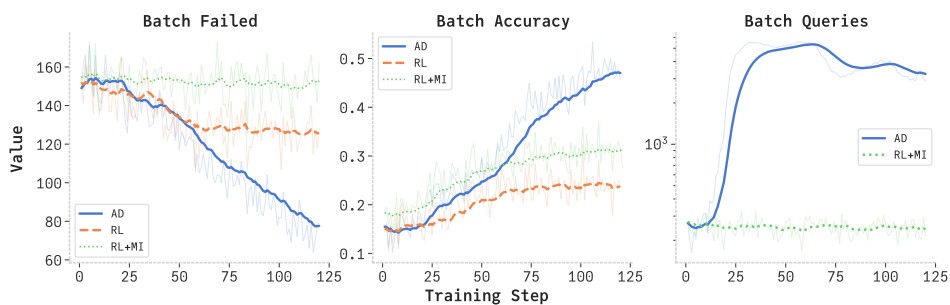

Figure 4: Training Dynamics of Reinforcement Learning, Reinforcement Learning with Masked Interaction, and AGENTIC DISTILLATION on Qwen2.5-7B-Instruct with Qwen3-30B-A3B-Instruct-2507 as the teacher LLM.

Table 2: Ablation study of AGENTIC DISTILLATION w.r.t. the modified importance sampling coefficient (abbreviated as IS) and clipping strategy (abbreviated as CS).

| Methods | Math | | | | Science | Code | | Puzzle |
|---|---|---|---|---|---|---|---|---|
| | AIME24 Avg@16 | AIME25 Avg@16 | MATH500 Avg@4 | LMB Avg@4 | GPQA-D Avg@4 | MBPP Avg@4 | LCB Avg@4 | RG Avg@4 |
| Student LLM: *Qwen2.5-7B-Instruct*, Teacher LLM: *Qwen3-30B-A3B-Instruct-2507* | | | | | | | | |
| AD | **14.82** | **14.33** | **78.17** | **14.27** | **37.13** | **62.73** | **18.62** | **21.11** |
| w/o IS | 13.44 | 13.56 | 77.25 | 14.11 | 36.82 | 62.32 | 18.44 | 20.92 |
| w/o CS | 14.34 | 13.92 | 76.53 | 13.22 | 35.89 | 61.46 | 17.33 | 19.08 |

strongest baseline in several instances. These results demonstrate that AGENTIC DISTILLATION not only excels in task-specific settings but also enables robust generalization across domains with distinct reasoning ability.

### 3.3 ABLATION STUDY

**Impact of Importance Sampling Coefficient in AGENTIC DISTILLATION.** To assess the necessity of the modified importance sampling coefficient for feedback tokens in AGENTIC DISTILLATION, as introduced in Equation (12), we compare its performance against the importance sampling coefficient used in the vanilla reinforcement learning algorithm, as shown in Table 2. The results demonstrate that the proposed modified importance sampling coefficient consistently outperforms the vanilla RL approach, confirming its critical role in enhancing AGENTIC DISTILLATION's effectiveness.

**Impact of Clipping Strategy in AGENTIC DISTILLATION.** Similarly, we evaluate the clipping strategy proposed in Equation (14). As illustrated in Table 2, removing the clipping strategy leads to a substantial decline in model performance across all tested scenarios. This indicates that the clipping strategy effectively mitigates issues such as gradient vanishing, thereby significantly improving the performance of the student LLM.

### 3.4 DOES THE LLM LEARN NEW KNOWLEDGE AND CAPABILITIES THROUGH AGENTIC DISTILLATION?

To validate and elucidate the learning outcomes of AGENTIC DISTILLATION, we analyze its training dynamics and the expansion of the knowledge boundary of the LLM.

Figure 5: Knowledge Boundary Expansion of AGENTIC DISTILLATION-trained Student LLM.

**Analysis of Training Dynamics.** Figure 4 illustrates the dynamics of three key metrics during the training of Qwen2.5-7B-Instruct, with Qwen3-30B-A3B-Instruct-2507 as the teacher LLM,

using Reinforcement Learning (RL), Reinforcement Learning with Masked Interaction (RL+MI), and AGENTIC DISTILLATION (AD). First, we examine the proportion of problems in a batch that the student LLM fails to solve across all rollouts (*Batch Failed*). Training with AGENTIC DISTILLATION significantly reduces this proportion, indicating that AGENTIC DISTILLATION enables the student LLM to acquire new knowledge, allowing it to solve previously unsolvable problems. This improvement is mirrored in the training batch accuracy, where AGENTIC DISTILLATION-trained LLMs show markedly higher gains. Additionally, we analyze the number of queries raised by the student LLM per batch (*Batch Queries*). With AGENTIC DISTILLATION, the number of queries initially increases, then decreases, and eventually stabilizes. This trend suggests that early in training, the student LLM queries the teacher LLM frequently to learn new knowledge. As its knowledge boundary expands, the student LLM relies less on queries, solving problems independently. These metric dynamics demonstrate that AGENTIC DISTILLATION effectively facilitates the student LLM's acquisition of new knowledge and capabilities from the teacher LLM.

**Knowledge Boundary Expansion of AGENTIC DISTILLATION-trained Student LLM.** To further study the change of the knowledge boundary of the AGENTIC DISTILLATION-trained student LLM, we collect the queries raised by student LLM Qwen2.5-7B-Instruct during the training process with Qwen3-30B-A3B-Instruct-2507 as the teacher LLM. We let the student LLM at different training stages to answer these queries and report the consistency with the answers of the teacher LLM. The consistency is judged by the GPT-4o (OpenAI, 2023). As shown in Figure 5, we can observe that as training progresses, the consistency of the student LLM on these unsolvable problems that are beyond its knowledge boundary gradually aligns with that of the teacher LLM, indicating that AGENTIC DISTILLATION can effectively inject the knowledge of the teacher LLM into the student LLM.

# 4 RELATED WORK

**Distilling from Strong LLMs.** Recent advancements in LLMs have led to remarkable performance in complex reasoning tasks (OpenAI, 2024a;b; 2025; DeepSeek-AI et al., 2025; Kimi-Team et al., 2025; Yang et al., 2025; Comanici et al., 2025; Huang & Yang, 2025; xAI, 2025). However, these models are often closed-source or possess an excessively large number of parameters, limiting their practical applications. Consequently, recent research has focused on distilling the capabilities of these strong reasoning LLMs into smaller-scale LLMs. Early studies (Qin et al., 2024; Guan et al., 2025; DeepSeek-AI et al., 2025) demonstrated that a small dataset generated by strong LRMs can significantly enhance the reasoning performance of smaller LLMs. Subsequent works (Bespoke-Labs, 2025; NovaSky-Team, 2025; Ye et al., 2025; Wen et al., 2025; Guha et al., 2025; Yang et al., 2025) have further improved distillation by optimizing problem set quality, curated data, training methods, loss functions, and integration of training stages. These approaches typically rely on distilling complete reasoning trajectories, a passive learning method that often fails to address the specific capabilities and knowledge gaps of student LLMs. In contrast, our proposed method enables student LLMs to actively query strong LRMs and selectively learn knowledge beyond their current capabilities, offering a more efficient and effective distillation.

**Enhancing LLM Reasoning with External Information.** Despite the remarkable performance of LLMs in various reasoning tasks, their capabilities are limited by inherent knowledge constraints and the fundamental limitations of deep learning architectures, which hinder their effectiveness in certain real-world tasks (Wang et al., 2025; Yang et al., 2024b). Prior work has employed reinforcement learning algorithms to enhance LLM decision-making, equipping them with autonomous capabilities such as planning, reasoning, tool usage, memory maintenance, and self-reflection (Wang et al., 2025; Singh et al., 2025; Jin et al., 2025; Liu et al., 2025). These efforts have improved LLM performance in knowledge-intensive question answering (Jin et al., 2025; Song et al., 2025), mathematical reasoning (Li et al., 2025; Bai et al., 2025), planning (Liu et al., 2025), and real-world applications (Mialon et al., 2024; Zhang et al., 2025b). However, these methods primarily focus on enhancing LLMs' ability to utilize tools to improve task performance. In contrast, we propose a distillation approach that leverages interactions between student LLMs and an external environment, specifically, teacher LLMs, to enhance reasoning capabilities without relying on the external information during inference.

## 5 CONCLUSION

In this paper, we introduce AGENTIC DISTILLATION, a distillation framework that enables active and adaptive knowledge transfer from strong LLMs to smaller student models. AGENTIC DISTILLATION leverages interaction and feedback tokens from teacher LLMs, allowing student models to selectively refine their reasoning and bridge knowledge gaps. To tackle off-policy and gradient vanishing issues inherent in learning from feedback, we introduce a tailored importance sampling coefficient and clipping strategy that seamlessly integrate into the reinforcement learning objective. Extensive experiments demonstrate that AGENTIC DISTILLATION achieves consistent improvements in both in-domain and out-of-domain reasoning tasks. We believe our framework could provide a promising direction for equipping compact models with advanced reasoning abilities.

ETHICS STATEMENT

This research focuses exclusively on LLM research problems and poses no risks to safety, personal security, or privacy. No new datasets are released as part of this study. Additionally, the research does not encompass potentially harmful insights, methods, or applications, nor does it raise issues related to privacy, security, legal compliance, or research integrity. We foresee no ethical risks or conflicts of interest. We are committed to adhering to ethical guidelines throughout the research process.

REPRODUCIBILITY STATEMENT

We provide a comprehensive description of the proposed AGENTIC DISTILLATION in § 2, with detailed implementation specifics provided in § A and § 3.1. All datasets utilized in this research are publicly available. Key code implementations are included in the supplementary materials for reference, and the complete code will be made publicly available upon acceptance of the paper.

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

# Appendix

## Table of Contents

# A  MORE IMPLEMENTATION DETAILS

## A.1  TRAINING DETAILS

Training utilized the veRL (Sheng et al., 2025) and vLLM (Kwon et al., 2023) frameworks on the clusters equipped with NVIDIA H100 GPUs. Table 3 present the detailed training parameters for AGENTIC DISTILLATION.

Table 3: Training Parameters.

| Parameters | Values |
|---|---|
| Batch Size | 256 |
| Number of Rollout Per Question | 8 |
| Rollout Temperature | 1.0 |
| Rollout Top-$p$ | 1.0 |
| Rollout Top-$k$ | -1 |
| Maximum Number of Generation Tokens | 16384 |
| Learning Rate | 1e-6 |
| KL Loss Coefficient | 0.001 |
| $\epsilon_{min}$ | 0.2 |
| $\epsilon_{max}$ | 0.28 |
| Gradient Clipping | 1.0 |
| Number of Training Steps | 300 |

## A.2  FULL TRAINING PROMPT

Prompt A.1 illustrates the full training prompt.

---

**Prompt A.1: Full Training Prompt**

OBJECTIVE:

To answer a User's question by providing a clear, verifiable reasoning process, potentially interacting with an external environment.

INTERACTION PROTOCOL:

For each question you receive, you MUST follow this two-step process:

**Step 1: Reasoning Process**

- **Decomposition:** Break down the user's question into a logical, step-by-step sequence of reasoning. Start from the most basic facts and build upon them.

- **External Inquiry (Optional but Encouraged):**
    - You may issue up to *max_turns* queries to an External Environment to validate hypotheses, clarify information, or advance your reasoning.
    - Each query must be a self-contained question enclosed in `<query>...</query>` tags.
    - **Wait for the `<result>...</result>` block** from the environment before continuing your reasoning.
    - **Critically analyze and integrate** the content from the `<result>...</result>` block into your reasoning chain.
    - Do not invent, assume, or hallucinate any `<result>` content. Your reasoning must be grounded in the provided results.

**Step 2: Final Answer**

- After your reasoning is complete, state your final answer clearly.
- The final answer, and only the final answer, MUST be enclosed in "\boxed{...}".

---

## A.3 INFERENCE PROMPT

To focus on distilling knowledge and capabilities from the teacher LLM, we prohibit the trained student LLM from interacting with the teacher LLM during the inference phase. For mathematical and puzzle reasoning benchmarks, we employ the prompt specified in Prompt A.2. For science and code reasoning benchmarks, we use the default prompts provided with the original benchmarks.

> **Prompt A.2: Prompt for Mathematical Reasoning Benchmarks**
>
> {question}
> Please reason step by step, and put your final answer within "\boxed{...}".

## A.4 TRAINING DATA

The training data of AGENTIC DISTILLATION is composed of three parts:

- **DAPO-Math-17K.** DAPO-Math-17K (Yu et al., 2025) is a dataset comprising 17,000 mathematical problems with integer answers, specifically designed for large-scale reinforcement learning of LLMs. The dataset was meticulously curated to ensure accurate reward signals by collecting questions and answers from the Art of Problem Solving (AoPS) website and competition homepages, followed by manual annotation and conversion to unify answers in integer form. We utilize the English subset, consisting of 14,000 questions, for training.

- **OpenScienceReasoning-2.** OpenScienceReasoning-2 is a multi-domain synthetic dataset aimed at enhancing general-purpose reasoning in LLMs. It includes multiple-choice and open-ended question-answer pairs with detailed reasoning traces, covering diverse scientific domains such as STEM, law, economics, and humanities. We randomly sample 20,000 examples from the original dataset for training.

- **Reasoning-Gym.** Reasoning-Gym (Stojanovski et al., 2025) is a community-developed Python library featuring procedural dataset generators and algorithmically verifiable reasoning environments for training reasoning models with RL. It encompasses over 100 tasks across domains including algebra, arithmetic, computation, cognition, geometry, graph theory, logic, and various games. We generate 27,000 samples for training, with each of 27 configurations producing 1,000 samples.

```
tasks = [
    ("ab", 1.0, {
        "seed": 42,
        "length": 10,
        "size": size
    }),
    ("ab", 1.0, {
        "seed": 42,
        "length": 15,
        "size": size
    }),
    ("acre", 1.0, {
        "seed": 42,
        "size": size
    }),
    ("advanced_geometry", 1.0, {
        "seed": 42,
        "min_coord": -100,
        "max_coord": 100,
        "size": size
    }),
    ("aiw", 1.0, {
        "seed": 42,
        "max_entities": 10,
        "size": size
    }),
    ("cryptarithm", 1.0, {
```

```
                        "seed": 42,
                        "min_words": 5,
                        "max_words": 20,
                        "size": size
                    }),
                    ("dice", 1.0, {
                        "seed": 42,
                        "num_dice": 5,
                        "max_dice_size": 30,
                        "size": size
                    }),
                    ("futoshiki", 1.0, {
                        "seed": 42,
                        "size": size
                    }),
                    ("game_of_life", 1.0, {
                        "seed": 42,
                        "grid_size_x": 30,
                        "grid_size_y": 30,
                        "simulation_steps": 3,
                        "size": size
                    }),
                    ("game_of_life", 1.0, {
                        "seed": 42,
                        "grid_size_x": 30,
                        "grid_size_y": 30,
                        "simulation_steps": 4,
                        "size": size
                    }),
                    ("game_of_life", 1.0, {
                        "seed": 42,
                        "grid_size_x": 30,
                        "grid_size_y": 30,
                        "simulation_steps": 5,
                        "size": size
                    }),
                    ("game_of_life_halting", 1.0, {
                        "seed": 42,
                        "grid_size_x": 30,
                        "grid_size_y": 30,
                        "difficulty": 3,
                        "num_oscillators": 8,
                        "max_simulation_steps": 40,
                        "size": size
                    }),
                    ("jugs", 1.0, {
                        "seed": 42,
                        "difficulty": 20,
                        "size": size
                    }),
                    ("knight_swap", 1.0, {
                        "seed": 42,
                        "size": size
                    }),
                    ("knights_knaves", 1.0, {
                        "seed": 42,
                        "n_people": 3,
                        "depth_constraint": 3,
                        "width_constraint": 3,
                        "size": size
                    }),
                    ("knights_knaves", 1.0, {
                        "seed": 42,
                        "n_people": 5,
                        "depth_constraint": 5,
```

```
1080                              "width_constraint": 5,
1081                              "size": size
1082                         }),
1083                         ("mahjong_puzzle", 1.0, {
1084                              "seed": 42,
1085                              "min_num_rounds": 30,
1086                              "size": size
1087                         }),
1088                         ("needle_haystack", 1.0, {
1089                              "seed": 42,
1090                              "min_num_statements": 50,
1091                              "size": size
1092                         }),
1093                         ("quantum_lock", 1.0, {
1094                              "seed": 42,
1095                              "difficulty": 10,
1096                              "size": size
1097                         }),
1098                         ("quantum_lock", 1.0, {
1099                              "seed": 42,
1100                              "difficulty": 20,
1101                              "size": size
1102                         }),
1103                         ("rush_hour", 1.0, {
1104                              "seed": 42,
1105                              "min_moves": 10,
1106                              "size": size
1107                         }),
1108                         ("self_reference", 1.0, {
1109                              "seed": 42,
1110                              "difficulty": 10,
1111                              "size": size
1112                         }),
1113                         ("sudoku", 1.0, {
1114                              "seed": 42,
1115                              "size": size
1116                         }),
1117                         ("zebra_puzzles", 1.0, {
1118                              "seed": 42,
1119                              "num_people": 4,
1120                              "num_characteristics": 4,
1121                              "size": size
1122                         }),
1123                         ("zebra_puzzles", 1.0, {
1124                              "seed": 42,
1125                              "num_people": 5,
1126                              "num_characteristics": 5,
1127                              "size": size
1128                         }),
1129                         ("zebra_puzzles", 1.0, {
1130                              "seed": 42,
1131                              "num_people": 6,
1132                              "num_characteristics": 6,
1133                              "size": size
                         }),
                         ("zebra_puzzles", 1.0, {
                              "seed": 42,
                              "num_people": 7,
                              "num_characteristics": 7,
                              "size": size
                         })
                    ]
```

## A.5 EVALUATION BENCHMARKS

The following details describe our evaluation benchmarks:

- **AIME24.** AIME24 comprises 30 challenging questions from the 2024 American Invitational Mathematics Examination (AIME), designed to test advanced mathematical reasoning skills.
- **AIME25.** AIME25 includes 30 challenging questions from the 2025 American Invitational Mathematics Examination (AIME), focusing on complex mathematical problem-solving.
- **MATH500.** The original MATH dataset (Hendrycks et al., 2021) contains 12,500 problems from American high school mathematics competitions. For this study, we use MATH500 (Lightman et al., 2024), a subset of the test split consisting exclusively of Level 5 questions.
- **LiveMathBench.** LiveMathBench (Liu et al., 2024b) is a continuously updated dataset of challenging mathematical problems. We utilize the December 2024 hard split, which includes 45 questions in English and Chinese.
- **GPQA.** The Graduate-Level Google-Proof Q&A Benchmark (GPQA) (Rein et al., 2023) is a challenging dataset of professional-level, multiple-choice science questions. We evaluate on its diamond subset, comprising 198 questions.
- **MBPP.** The Mostly Basic Programming Problems (MBPP) dataset (Austin et al., 2021) evaluates programming models on basic Python tasks. Constructed via crowdsourcing, the problems and solutions undergo revision and manual inspection to ensure clarity and accurate test cases.
- **LiveCodeBench.** LiveCodeBench (Jain et al., 2025) is a benchmark for comprehensive and uncontaminated evaluation of LLM code-related capabilities, incorporating questions from LeetCode, AtCoder, and Codeforces.
- **Reasoning-Gym.** Using the configurations outlined in Appendix A.4, we generate 270 samples for evaluation, with each of 27 configurations producing 10 samples.

## B ADDITIONAL EXPERIMENTAL RESULTS AND ANALYSIS

### B.1 PERFORMANCE OF TEACHER LLMS

Table 4 shows the performance of Qwen3-30B-A3B-Instruct-2507 and GPT-OSS-20B, which are utilized as teacher LLMs in this work. The performance of teacher LLMs can be seen as the upper bound of the distillation.

Table 4: Performance of Qwen3-30B-A3B-Instruct-2507 and GPT-OSS-20B.

| Methods | Math | | | | Science | Code | | Puzzle |
| --- | --- | --- | --- | --- | --- | --- | --- | --- |
| | AIME24 Avg@16 | AIME25 Avg@16 | MATH500 Avg@4 | LMB Avg@4 | GPQA-D Avg@4 | MBPP Avg@4 | LCB Avg@4 | RG Avg@4 |
| *Qwen3-30B-A3B-Instruct-2507* | | | | | | | | |
| - | 76.88 | 63.96 | 96.75 | 44.50 | 55.18 | 84.05 | 44.74 | 19.54 |
| *GPT-OSS-20B* | | | | | | | | |
| - | 78.62 | 73.75 | 96.45 | 50.50 | 59.22 | 93.68 | 60.53 | 13.98 |

### B.2 AGENTIC DISTILLATION ON LARGER STUDENT LLMS

In this section, we evaluate the effectiveness of AGENTIC DISTILLATION on student LLMs with larger parameter sizes, specifically training Qwen2.5-32B-Instruct (Yang et al., 2024a) with AGENTIC DISTILLATION. As shown in Table 5, the evaluation results demonstrate that AGENTIC DISTILLATION remains effective for larger-scale models, with AGENTIC DISTILLATION-trained models outperforming baseline models across all benchmarks. Notably, the performance improvements for Qwen2.5-32B-Instruct are more pronounced compared to those for Qwen2.5-7B-Instruct. This enhanced improvement may stem from the 32B model's stronger baseline capabilities, enabling it to formulate higher-quality questions and acquire knowledge more efficiently during training with AGENTIC DISTILLATION.

Table 5: Experimental results of AGENTIC DISTILLATION and baselines with Qwen2.5-32B-Instrcut as the student LLM. We report the average performance for 16 runs on AIME24 and AIME25, and 4 runs on others. We abbreviate LMB as LiveMathBench v202505, LCB as LiveCodeBench v6, RG as Reasoning Gym, MI as Masked Interaction, and AD as AGENTIC DISTILLATION.

| Methods | Math | | | | Science | Code | | Puzzle |
|---|---|---|---|---|---|---|---|---|
| | AIME24 Avg@16 | AIME25 Avg@16 | MATH500 Avg@4 | LMB Avg@4 | GPQA-D Avg@4 | MBPP Avg@4 | LCB Avg@4 | RG Avg@4 |
| Student LLM: *Qwen2.5-32B-Instruct*, Teacher LLM: *Qwen3-30B-A3B-Instruct-2507* | | | | | | | | |
| Original | 14.38 | 13.12 | 80.85 | 12.25 | 47.10 | 85.12 | 24.71 | 13.06 |
| +SFT | 16.83 | 15.79 | 83.47 | 15.33 | 46.13 | 84.74 | 23.15 | 15.61 |
| +RL | 15.73 | 14.11 | 82.51 | 13.33 | 50.89 | 86.71 | 26.43 | 17.62 |
| +AD | **17.56** | **18.19** | **86.82** | **17.37** | **51.05** | **88.74** | **26.91** | **22.48** |

## B.3 AGENTIC DISTILLATION ON LONG-COT STUDENT LLMS

In this section, we assess the performance of AGENTIC DISTILLATION on reasoning LLMs utilizing long CoT prompting. Given the substantial inference overhead of long CoT LLMs, we conducted experiments using DeepSeek-R1-Distill-Qwen-1.5B, with results presented in Table 6. The findings demonstrate that AGENTIC DISTILLATION achieves consistent performance improvements for student LLMs with extended reasoning chains, underscoring the generalization capability of AGENTIC DISTILLATION across such models.

Additionally, we observe a performance decline in models trained with SFT. This may be attributed to the teacher LLM, Qwen3-30B-A3B-Instruct-2507, not being optimized for long CoT reasoning. Consequently, fine-tuning based on its responses may disrupt the original reasoning patterns of the student LLM, leading to degraded performance. In contrast, AGENTIC DISTILLATION selectively injects knowledge into the student LLM via query-answer pairs, preserving its inherent reasoning patterns. This preservation represents a key advantage of AGENTIC DISTILLATION, enhancing its effectiveness without compromising the student LLM's original reasoning capabilities.

Table 6: Experimental results of AGENTIC DISTILLATION and baselines with DeepSeek-R1-Distill-Qwen-1.5B as the student LLM. We report the average performance for 16 runs on AIME24 and AIME25, and 4 runs on others. We abbreviate LMB as LiveMathBench v202505, LCB as Live-CodeBench v6, RG as Reasoning Gym, MI as Masked Interaction, and AD as AGENTIC DISTILLATION.

| Methods | Math | | | | Science | Code | | Puzzle |
|---|---|---|---|---|---|---|---|---|
| | AIME24 Avg@16 | AIME25 Avg@16 | MATH500 Avg@4 | LMB Avg@4 | GPQA-D Avg@4 | MBPP Avg@4 | LCB Avg@4 | RG Avg@4 |
| Student LLM: *DeepSeek-R1-Distill-Qwen-1.5B*, Teacher LLM: *Qwen3-30B-A3B-Instruct-2507* | | | | | | | | |
| Original | 21.88 | 21.46 | 83.95 | 13.00 | 29.80 | 60.12 | 14.69 | 3.33 |
| +SFT | 18.35 | 19.89 | 77.16 | 14.02 | 26.64 | 55.51 | 15.27 | 10.98 |
| +RL | 28.43 | 25.70 | 86.82 | 17.39 | 34.68 | 65.05 | 14.72 | 13.53 |
| +AD | **30.56** | **29.21** | **88.47** | **18.90** | **36.53** | **67.19** | **17.06** | **16.44** |

## B.4 CASE STUDY

We present examples of AGENTIC DISTILLATION's training trajectories in Case B.1 and Case B.2. These instances demonstrate the student model's ability to formulate reasonable and meaningful queries, leverage answers from the teacher model, and successfully complete the reasoning process. Through this interaction, the student effectively acquires new information and enhances its inherent capabilities.

Case B.1: Case on AIME2024

## QUESTION

Jen enters a lottery by picking 4 distinct numbers from $S = \{1, 2, 3, \cdots, 9, 10\}$. 4 numbers are randomly chosen from $S$. She wins a prize if at least two of her numbers were 2 of the randomly chosen numbers, and wins the grand prize if all four of her numbers were the randomly chosen numbers. The probability of her winning the grand prize given that she won a prize is $\frac{m}{n}$ where $m$ and $n$ are relatively prime positive integers. Find $m + n$.

## ANSWER

116

## TRAJECTORY

**Step 1: Reasoning Process**

- **Decomposition:**

  1. **Identify the Objective:** Find the sum $m + n$, where $\frac{m}{n}$ is the conditional probability that Jen wins the grand prize given that she wins *any* prize.
  2. **Define the Events:**
     - Total pool of numbers: $S = \{1, 2, \ldots, 10\}$ (Size 10).
     - Jen's pick ($J$): 4 specific numbers.
     - Winning numbers ($W$): 4 randomly chosen numbers.
     - Event $E$ (Prize): The intersection $|J \cap W|$ is at least 2 (i.e., 2, 3, or 4 matches).
     - Event $G$ (Grand Prize): The intersection $|J \cap W|$ is exactly 4.
  3. **Set up the Formula:** The conditional probability is $P(G|E) = \frac{N(G \cap E)}{N(E)}$. Since $G$ is a subset of $E$ (4 matches implies at least 2), this simplifies to $\frac{N(G)}{N(E)}$.
  4. **Combinatorial Model:** To find the number of ways to match exactly $k$ numbers:
     - We need to choose $k$ numbers from the 4 winning numbers: $\binom{4}{k}$.
     - We need to choose the remaining $4 - k$ numbers from the 6 non-winning numbers: $\binom{6}{4-k}$.
  5. **External Verification:** I need to calculate the specific combinatorial values for $k = 2, 3, 4$ and their sum to determine $N(E)$ and $N(G)$. I will ask an external environment to perform these calculations.

- **External Inquiry:**
  ```
  <query>
  ```
  Calculate the number of ways to match exactly k numbers in a lottery where 4 numbers are drawn from 10. Specifically, compute the values for C(4,k) * C(6, 4-k) for k=2, k=3, and k=4, and find the sum of these three results.
  ```
  </query>
  ```
  ```
  <result>
  ```
  115
  ```
  </result>
  ```

- **Analysis and Integration:**

  - The external result confirms the counts for each case.
  - $N(E)$ **(Total ways to win a prize):** This is the sum of outcomes for $k = 2, 3, 4$. According to the result, $90 + 24 + 1 = 115$.
  - $N(G)$ **(Ways to win grand prize):** This corresponds to the $k = 4$ case, which is 1.
  - **Calculate Probability:** The conditional probability is $\frac{1}{115}$.
  - **Identify $m$ and $n$:** $m = 1$, $n = 115$.
  - **Check Condition:** 1 and 115 are relatively prime positive integers.
  - **Final Step:** Calculate $m + n = 1 + 115 = 116$.

**Step 2: Final Answer**

The probability that Jen wins the grand prize given she wins a prize is $\frac{1}{115}$. Therefore, $m = 1$ and $n = 115$, and the sum $m + n$ is $\boxed{116}$.

---

**Case B.2: Case on Bamboogle**

QUESTION

What is the capital of the second largest state in the US by area?

ANSWER

austin

TRAJECTORY

**Step 1: Reasoning Process**

- **Decomposition:** To determine the capital of the second largest state in the US by area, I need to follow a logical sequence.
    - First, I must definitively identify which US state ranks second in terms of total area.
    - Second, once that state is identified, I need to determine its capital city.
- **External Inquiry:** I will start by asking the external environment to identify the second largest state in the US.

    ```
    <query>
    What is the second largest state of US?
    </query>
    ```

    ```
    <result>
    Texas
    </result>
    ```

    The external result confirms that **Texas** is the second largest state in the US by area. Now I need to find the capital of Texas.

    ```
    <query>
    What is the capital of Texas?
    </query>
    ```

    ```
    <result>
    Austin
    </result>
    ```

    From the retrieved information, I have verified that the capital of Texas is Austin.

**Step 2: Final Answer**

- The second largest state in the US by area is Texas, and its capital is Austin.
- $\boxed{Austin}$

---

## C    LIMITATIONS

In this paper, we propose AGENTIC DISTILLATION, a novel framework for active distillation to enhance the reasoning capabilities of LLMs. While AGENTIC DISTILLATION achieves significant performance improvements, several areas warrant further exploration. First, our framework does not explicitly guarantee the accuracy of feedback provided to the student LLM. In cases where the teacher produces suboptimal or noisy guidance, the student may inadvertently learn misleading patterns, which could diminish the effectiveness of the distillation process and lead to unstable improvements across tasks. Second, as the scale and inference complexity of teacher LLMs increase, the training time required by AGENTIC DISTILLATION may grow considerably. This not only elevates computational costs but may also impose practical challenges when deploying the framework in resource-constrained environments or when scaling to very large datasets and extended training regimes.

# D LLM USAGE

In this paper, the use of LLMs is intentionally restricted to the final stages of the research process, specifically for refining and proofreading the written content. The LLMs are employed solely to enhance the clarity, coherence, and grammatical accuracy of the text, ensuring effective and professional communication of the presented ideas. Importantly, LLMs played no role in the core components of this work, including the development of the research methodology, the design of experiments, or the analysis of results. We are aware that we will be responsible for all content in the paper.

