# OpenReview forum: "Learning with Interaction: Agentic Distillation for Large Language Model Reasoning"
_ICLR.cc/2026/Conference — ICLR 2026 Conference Withdrawn Submission_

### Official Review · Reviewer_qKrD · 2025-10-24

**Soundness:** 3
**Presentation:** 2
**Contribution:** 3
**Rating:** 4
**Confidence:** 4

**Summary:**

This paper introduces Agent Distillation to address existing data-centric distillation, which over learn on easy samples. They propose letting a student model interact with a stronger teacher model, and ask for the teacher's feedback when the student is not able to solve the problem on their own. This feedback will be used to guide the student, and it shows that learning from teacher-generated feedback effectively improves distillation performance.

**Strengths:**

1. The detailed discussion of several issues when trying to inject teacher-generated tokens into the student LM is insightful (e.g., being off-policy and gradient vanishing), and the author provides solutions to address these issues.
2. The evaluation is conducted on extensive tasks, and the benchmarks are well chosen.
3. The improvement is consistent across tasks, showing strong performance from learning from the teacher’s feedback.

**Weaknesses:**

1. Some qualitative analysis will help, for example, show what the student is actually generating after training, and how it improves the performance. Does the student also generate feedback-style reasoning during test time?
2. Adding a baseline on SFT from the teacher’s full trajectory (including the feedback) and then doing RL for correctness would further strengthen the claim. How important is the interaction? Can we collect feedback in an offline manner?
3. The method figure is not clear; for example, it's hard to see that the teacher is generating multi-turn feedback. Also, how to decide when the student model needs help is not clear either.
4. "When to use external feedback" is controlled solely by prompting the student model. However, models can be over-confident or ill-calibrated, necessitating the need for an analysis on how often the student is over-confident (does not call the teacher model but cannot solve the problem on its own).
5. Some variants on when to use external feedback are also necessary to justify this design choice.

**Questions:**

1. Typo in Fig 1’s caption: gap instead of grap.
2. Text in Fig 3 is too small, especially for the equations.
3. There is an additional a’ in line 119.
4. The teacher feedback is used when any single rollout fail, or when all rollouts fail for a question? If it’s the former, how does this method avoid overlearning, since the student model is actually able to solve this question (but not always correct).
5. Using the objective from equation 16, does the student model also generate self-correct or feedback-style content during test time?
6. What if the SFT baseline is also distilling teacher’s feedback (and followed by RL training for correctness)? What would the performance be like
7. In abstract, the statement "...reasoning abilities to solve complex tasks, which has propelled the progress toward artificial general intelligence" is not necessary or should be backed by citations.
8. In the Knowledge Boundary Expansion analysis, which dataset is this, and what is the sample size? In my opinion, the more direct way to test this is to see how many previously unsolvable problems become solvable after training.

Missing reference on distilling from interaction: https://arxiv.org/abs/2402.01620

---

> ### Author Response · Authors · 2025-11-19
> **Response to Reviewer qKrD**
>
> Dear reviewer qKrD,
>
> Thanks for recognizing our insightful idea, extensive experiments, and consistent improvement. Below, we address the concerns you raised.
>
> > Response to W1: Qualitative analysis and test-time generation
>
> We will include additional case studies in the final version. The student does **not** generate feedback-style interactions at test time. It uses a standard Chain-of-Thought format. The training process expands the student's "Knowledge Boundary" (Figure 5), allowing it to solve problems autonomously that it previously could not.
>
> > Response to W2: Baseline of SFT on full teacher trajectory
>
> We included this baseline, denoted as **"RL+SFT"** in Table 1. This baseline uses the teacher's data (SFT) followed by RL for correctness. Agentic Distillation (AD) consistently outperforms "RL+SFT" (e.g., 14.82 vs 12.13 on AIME24). This confirms that the *active* nature of AD, where the student queries based on its specific uncertainty, is superior to passive SFT on full trajectories.
>
> > Response to W3 & W4: Mechanism of "When to ask" and Overconfidence
>
> - **Mechanism:** The student decides when to ask. The policy $\pi_\theta$ generates the `<query>` token autonomously during the rollout.
> - **Overconfidence:** The RL reward mechanism naturally calibrates confidence. If the student fails to ask when necessary, it receives a low reward (incorrect answer). If it asks effectively, it gets a high reward. **Figure 4** shows "Batch Failed" decreasing, indicating the model learns the optimal balance of when to use external help vs. internal reasoning.
>
> > Response to W5: Some variants on when to use external feedback are also necessary to justify this design choice.
>
> The student LLM will continue reasoning based on the external feedback corresponding to the previously proposed query. Therefore, the external feedback is appended after the query, and using external feedback in other stages will hinder the student LLM's continued reasoning.
>
> > Response to Q1 & Q2 & Q3 & Q7: Some typos
>
> Thanks for your suggestions. We have fixed them in the new version.
>
> > Response to Q4: Teacher feedback usage
>
> Teacher feedback is used whenever the student generates a query. Even if the student *could* have solved it (but might have hallucinated), the feedback serves as a "guardrail" or correction mechanism during training, reinforcing accurate reasoning trajectories.
>
> > Response to Q5: Test time generation
>
> We utilize prompts to control whether the student LLM generates feedback-style content. During testing, the student LLM is prompted with the pure reasoning prompt described in Appendix A.3 and is prohibited from interacting with the teacher LLM. Without the instruction to interact, the student LLM will not generate feedback-style content.
>
> > Response to Q6: SFT Baselines
>
> Thank you for your suggestion. We provide the RL baseline initiated by SFT, as shown in the table below. Its performance is higher than that of pure SFT or RL, but it is still weaker than AgenticDistillation.
>
> |  | **AIME24** | **GPQA** | **LCB** |
> | --- | --- | --- | --- |
> | Original | 9.79 | 33.33 | 15.71 |
> | RL w/ SFT start | 12.27 | 35.91 | 16.59 |
> | AD | **14.82** | **37.13** | **18.62** |
>
> > Response to Q8: Details of Knowledge Boundary Expansion analysis
>
> In this paragraph, we aim to further verify whether the proposed AgenticDistillation can expand the knowledge boundary of the student LLM. The way you mentioned is indeed a feasible solution, which is already reflected in **Figure 4** and **Table 1**. Here, we aim to further corroborate this through the queries proposed by the student LLM during the training process. As described in Lines 439-440, we collected the set of queries proposed by Qwen2.5-7B-Instruct during the training process, sampled approximately 10k queries from it, and then used GPT-4o as a verifier to determine whether the answers of the student LLM at different steps to these queries are consistent with those of the teacher LLM, thereby verifying whether the student LLM has learned new knowledge from the teacher LLM through the feedback from the teacher LLM.
>
> ---
>
> Finally, thank you again for your detailed review. We hope that our responses could address your concerns. If you have any other questions, please don't hesitate to let us know.

---

### Official Review · Reviewer_3uKu · 2025-10-30

**Soundness:** 2
**Presentation:** 2
**Contribution:** 2
**Rating:** 2
**Confidence:** 4

**Summary:**

This paper introduces AgenticDistillation, a knowledge distillation (KD) approach. The authors motivate the idea with two problems in standard KD: overlearning (overfitting to simple questions) and knowledge gaps between the teacher and student. Their method prompts the student to actively seek teacher feedback on particular steps and allows models to learn from the feedback the teacher provides. The paper then introduces a gradient clipping method for performing RL on and SFT on different parts of the response (question and response). The method is tested on datasets from math, code, and science domains, with improvements over the base models and ablations that they compare against

**Strengths:**

- Framing distillation as a method where the agent asks for information from an oracle is an interesting idea
- RL approach for learning from feedback appears novel
- The authors tested a variety of models, including reasoning and non-reasoning models.

**Weaknesses:**

- Under-reported baselines: In Table 1, the baselines the authors report seem lower than what has been reported in published work. For example, the Qwen 2.5 tech report (https://arxiv.org/pdf/2409.12122v1, Table 5) has AIME 24 performance at 5/30 (16%) while this paper reports 9%. Past work (e.g. https://arxiv.org/abs/2506.11902, Table 1) has also reported higher baseline numbers for MATH-500 (76.5%, rather than 73.00 reported here). In several cases, the gain reported from distillation largely disappears when considering the stronger baseline numbers. Can the authors explain why their baselines are consistently lower than prior work?

- No external baselines: All baselines compared against are internal models, but no other competing distillation methods were evaluated (e.g. https://arxiv.org/abs/2503.07067, https://aclanthology.org/2025.acl-industry.4/, https://arxiv.org/abs/2509.25837) although several are cited in related work.

- Potential data leakage: how sure are the authors that none of the datasets tested on are included in the training data sourced from DAPO, OpenScienceReasoning, and Reasoning Gym?

- It's not clear to me what happens at test time. During training, the model asks for information from the oracle -- is this prompt also followed at test time? If not, why is the model improving from training?

**Questions:**

# Comments
- Small quibble: I find the motivation w.r.t. AGI unnecessary, and a bit of a leap (i.e. the connection between distillation and AGI is a bit tenuous)
- many of the sentences are incomplete and the paper could use more polishing (e.g. L061)
- L324 typo in Instruct

---

> ### Author Response · Authors · 2025-11-19
> **Response to Reviewer 3uKu**
>
> Dear reviewer 3uKu,
>
> Thanks for recognizing our interesting and novel idea and extensive experiments. We respond to your concerns in the following.
>
> > Response to W1: Baseline reporting (lower than tech reports)
>
> To ensure a fair and rigorous comparison, we re-evaluated **all** models (baselines and our method) on our own cluster under identical settings. Discrepancies with technical reports arise because:
>
> - **Metric Definition:** Technical report of Qwen 2.5 uses `pass@1`, whereas we report the **average** performance over multiple runs to reduce variance. Similarly, the report in https://arxiv.org/abs/2506.11902 is based on the average of 3 times, while our experiment reports the average of 4 times. **According to previous work[1], the score after averaging multiple predictions will be lower but more reflective of the true capability.**
> - **Sampling Parameters:** We utilize specific temperature and top-p settings (detailed in Section 3.1) that may differ from the highly tuned settings in tech reports.
> - **Dataset Size:** AIME24 has only 30 questions; a single error can shift the score by ~3.3%, causing high variance.
>
> We believe our relative improvements hold significant weight because the control variables were strictly identical across all experiments.
>
> > Response to W2: Lack of external distillation baselines
>
> Thank you for your suggestions, but these papers basically follow the idea of traditional KD, which relies on the KL divergence between student LLMs and teacher LLMs as the objective function for distillation. However, this approach has significant limitations: **they require student LLMs and teacher LLMs to have the same tokenizer and vocabulary; otherwise, if the dimensions of the output logits are different, the calculation of KL divergence cannot be performed**.
>
> In the experimental setup of this paper, both student LLMs and teacher LLMs are from different LLM families, so we only compared baselines that use natural language as a bridge.
>
> Distillation methods that use natural language as a bridge also have higher applicability. Because when teacher LLMs are closed-source models or from different LLM families, methods based on KL divergence cannot work.
>
> > Response to W3: Data leakage concerns
>
> We strictly enforced decontamination:
>
> - **DAPO & OpenScience:** These are standard training sets constructed to exclude mainstream evaluation benchmark questions.
> - **Reasoning Gym:** This dataset is procedurally generated. We generated distinct seeds/configurations for training and testing to ensure zero overlap.
>
> > Response to W4: Test time behavior
>
> Student LLMs only ask teacher LLMs questions that are beyond their own knowledge boundaries during training. Then, through the policy optimization of RL, student LLMs can learn new knowledge from the answers of teacher LLMs. During testing, student LLMs are not allowed to interact with teacher LLMs, instead, they are prompted with typical reasoning prompts (as described in **Appendix A.3**).
>
> > Response to Questions
>
> Thanks for your suggestions. We have fixed them in the new version.
>
> [1] Liu, Junnan, et al. Are Your LLMs Capable of Stable Reasoning? *ACL 2025*.
>
> ---
>
> Thank you for your diligent review. We hope our reply can address your questions and concerns, and we are also open to potential other issues.

---

> > ### Comment · Reviewer_3uKu · 2025-11-26
> > **Response to rebuttal**
> >
> > Thanks for your response -- in order:
> >
> > > Baseline reporting
> >
> > While I understand that there can be differences in performance due to the reasons listed, it is incumbent on the authors to start from the strongest-possible starting point. Like I mention in my review, my concern is that in some cases the relative gains disappear when considering the stronger baseline numbers. In other words: If I can get equally-strong performance just  by changing the decoding parameters, why should I bother with implementing this method? An easy way to answer this question is to show improvements on top of the best set of parameters, which this work fails to do.
> >
> > >  Lack of external distillation baselines
> >
> > While I understand the tokenizer-based argument the authors make, it does not explain why they don't consider text-based distillation methods e.g. ones based on https://arxiv.org/abs/2110.07178, like https://arxiv.org/abs/2306.14050, https://arxiv.org/abs/2212.08410, etc. or other newer work that does "Distillation methods that use natural language as a bridge " which is a well-explored area.
> >
> > Due to these primary concerns, I will maintain my score.

---

> > > ### Author Response · Authors · 2025-11-27
> > >
> > > Thank you for your reply. Below is our supplementary response.
> > >
> > > > Baseline reporting
> > >
> > > Our study ensures a fair comparison by evaluating both our method and the baselines on identical backbone LLMs, thereby isolating the relative performance gains attributed specifically to our approach. Furthermore, we conducted experiments across diverse student-teacher pairs (using distinct backbone architectures, i.e., Qwen & Llama), demonstrating the generalization capabilities and transferability of our method to stronger models.
> > >
> > > Regarding the concern that `'get equally-strong performance just by changing the decoding parameters'` we argue that our method provides an improvement that is orthogonal to the base model's strength. Even with a stronger backbone or optimized decoding settings, applying our method would yield further performance gains. Since the definition of `'the best set of parameters'` can be ambiguous, we adhered to standard settings for widely-used open-source LLMs like Qwen and Llama. If there is a specific backbone or parameter configuration you have in mind, we would be grateful if you could specify it, and we will happily conduct further comparisons.
> > >
> > > > Lack of external distillation baselines
> > >
> > > Thanks for pointing out these works. Our choice of baselines is intentional, aiming to compare against the most representative paradigms. The works you mentioned operate on the principle of training the student model on teacher-generated CoT data. Since this fundamentally falls under the methodology of **Supervised Fine-Tuning (SFT) with data filter** (described in **Line 291-292**), we believe our current baselines provide a fair and sufficient comparison for assessing the effectiveness of our method.
> > >
> > > ---
> > >
> > > We hope that our response can address your concerns. The core contribution we aim to present is the **consistently relative improvement** of AgenticDistillation over **representative distillation methodology**. This motivated us to verify our method across **various base LLMs** and **representative distillation paradigms**

---

### Official Review · Reviewer_fuv7 · 2025-11-01

**Soundness:** 2
**Presentation:** 2
**Contribution:** 3
**Rating:** 6
**Confidence:** 4

**Summary:**

The paper proposes Agentic Distillation as a novel paradigm for knowledge transfer, aiming to distill complex reasoning capabilities from large, computationally expensive teacher models into smaller student models. Unlike traditional static knowledge distillation, this approach introduces an interactive and agent-based learning environment. The authors motivate this work by pointing out that current data-centric distillation methods suffer from passive learning, over-fitting on simple examples, and persistent knowledge gaps. While conceptually innovative, the method relies on dynamic interaction, which introduces non-trivial overhead and stability risks that must be comprehensively addressed.

**Strengths:**

1. The core idea of shifting from passive data-centric distillation (e.g., logit-matching) to an active, interactive, agent-based learning framework is a major intellectual contribution. It offers a genuine new direction for solving the knowledge gap problem.

2. The agentic distillation framework provides a flexible structure that could potentially integrate advanced components, such as tool-use or external memory, making the overall distillation process more comprehensive and future-proof.

**Weaknesses:**

1. The method contradicts its primary goal of efficiency by introducing significant training-phase complexity. The paper explicitly notes that training time "may grow considerably" with teacher complexity. This dramatically limits the ability of the deep learning community to reproduce, scale, or even test this method without substantial, often inaccessible, compute resources.

2. The success of the distillation is likely critically dependent on the specific design of the "interaction" protocol, the reward signals, and the complexity of the agent architecture. If the results are highly sensitive to these hyper-parameters, the methodology is not broadly applicable or robust.

3. Traditional distillation offers a clear, convex optimization target (e.g., KL divergence). Introducing complex, nested optimization loops and dynamic feedback makes the objective function non-trivial, harder to analyze, and obscures which components (the distillation loss, the agentic feedback, or the interaction environment) are providing the primary performance gains.

**Questions:**

The paper notes a risk of "unstable improvements across tasks" due to the dynamic, interactive nature of the training. Given that a key strength is the potential for generalized reasoning, how do the authors explicitly decouple the emergent reasoning skills from the specifics of the training interaction protocol? Specifically, how is the learned policy guaranteed to be a generalized reasoning model, rather than an agent that has merely overfit to the teacher's interactive prompt-response and self-correction style within the training environment, leading to a catastrophic collapse in performance on static, zero-shot benchmarks where the interactive scaffold is absent?

---

> ### Author Response · Authors · 2025-11-19
> **Response to Reviewer fuv7**
>
> Dear reviewer fuv7,
>
> Thank you very much for recognizing that our framework is a major intellectual contribution and is future-proof. Below, we will respond to your concerns.
>
> > Response to W1: Training complexity and efficiency
>
> We agree that interactive training incurs a computational cost, which we discuss in the **Limitations (Appendix C)** section. However, we emphasize two points regarding efficiency:
>
> 1. **Inference Efficiency:** The primary goal of distillation is to produce a small, efficient *student* model for deployment. Once trained, our student model performs standard Chain-of-Thought (CoT) reasoning **without** needing the teacher or oracle interactions.
> 2. **Reduced Teacher Inference:** Compared to methods that require generating full trajectories from the teacher for every training instance (Rejection Sampling), our method allows the student to query selectively. The student often solves easy steps independently, reducing the total token load required from the massive teacher model.
>
> > Response to W2: Robustness of the interaction protocol
>
> Our interaction protocol is designed to be simple and general. It relies on standard natural language prompting (enclosing queries in `<query>` tags) rather than complex, architecture-specific engineering. We empirically demonstrated robustness in **Section 3.2** and **Table 1**, showing consistent improvements across different student families (Qwen, Llama) and different teacher types (Long-CoT vs. Short-CoT). In **Section 3.3**, we also demonstrate that our method is not sensitive to several important hyperparameters.
>
> > Response to W3: Objective function analysis (RL vs. KD)
>
> Traditional distillation methods usually need to optimize two objectives simultaneously: KL divergence and correctness, which increases the risk of conflicting with each other.
>
> However, the method in this paper essentially has only one supervisory signal, which is the correctness reward. From this perspective, the method in this paper has a clearer optimization objective, and referring to the previous study of RL [1][2], the optimization objective of RLVR has a verified convergence.
>
> Regarding the source of primary performance gain, from the comparison results with RL+SFT and RL+MI, agentic feedback plays a primary role.
>
> > Response to Q1: Risk of overfitting to interaction (Generalization)
>
> This is a critical point. We address the risk of the model becoming dependent on the "interactive scaffold" by evaluating it on **standard, zero-shot benchmarks** where no interaction is allowed (prompt is detailed in Appendix A.3).
>
> - **Evidence (Table 1):** The student model shows significant gains on **out-of-domain** benchmarks (e.g., MBPP, LiveCodeBench) and static benchmarks (MATH500). This proves the model has internalized the *reasoning capability*, rather than overfitting to the *interaction format*.
>
> [1] Schulman, John, et al. Proximal policy optimization algorithms. *arXiv preprint 2017*.
>
> [2] Schulman, John, et al. Trust region policy optimization. *ICML 2015*.
>
> ---
>
> Thank you for your diligent review. We hope our reply addresses your questions and concerns, and we are open to any other discussions.

---

### Official Review · Reviewer_8BPw · 2025-11-06

**Soundness:** 2
**Presentation:** 2
**Contribution:** 2
**Rating:** 2
**Confidence:** 4

**Summary:**

The authors propose Agentic Distillation, a distillation method wherein a student LLM optionally queries a teacher LLM, in the process obtaining feedback which is then jointly optimized with its own generated tokens using GRPO. To stably learn from the teacher's feedback tokens (sampled from the teacher's policy), the authors introduce an importance sampling coefficient and a clipping strategy. Experiments are conducted on different reasoning and coding benchmarks with multiple student+teacher combinations to show that Agentic Distillation outperforms SFT on teacher trajectories and RL with its own trajectories (w/o any teacher interaction).

**Strengths:**

* In off-policy distillation, a student passively learns from teacher trajectories instead of learning from the feedback obtained on its own (student's) trajectories. Agentic Distillation presents a way of mitigating this issue.

* Experiments are pretty thorough with the main result being that actively learning from teacher's feedback tokens can be more beneficial than imitating teacher's trajectories.

**Weaknesses:**

* An important missing baseline is on-policy distillation. It is the most common and effective way of distillation which has been shown to outperform off-policy distillation. It is also compute-efficient because querying the teacher’s log probabilities requires just a single forward pass from the larger model, while the trajectories are generated by the smaller and cheaper student.

* The paper lacks examples and analysis of the kind of queries that the student generates for the teacher and the teacher's subsequent feedback. Without such analysis, it's hard to understand if the student only asks for hints or full answers? Additionally, what is stopping the teacher from giving out complete answers in which case, agentic distillation will turn into vanilla off-policy distillation. In summary, I'm unsure how the student learns to balance between always asking the teacher for complete solutions versus never interacting. Even though the authors write "This trend suggests that early in training, the student LLM queries the teacher LLM frequently to learn new knowledge.", this requires more analysis, examples, and explanation.

[1] On-Policy Distillation of Language Models: Learning from Self-Generated Mistakes. Agarwal et al., 2023

**Questions:**

* How does your method compare to on-policy distillation?

* "This method employs a temperature coefficient to sharpen the teacher LLM’s distribution" -- Could you explain this more? How do you use the temperature?

* Did you try experimenting with a weaker teacher and a stronger student?

* Since your teacher LLMs are thinking LLMs, does the student also learn from the think tokens or only the response tokens?

* Can you share some examples and statistics of the queries generated by the student? Similarly, the teacher's feedback would also be interesting to look at.

---

> ### Author Response · Authors · 2025-11-19
> **Response to Reviewer 8BPw**
>
> Dear reviewer 8BPw,
>
> Thank you for taking the time to review our work; we will address your concerns and doubts below.
>
> > Response to W1 & Q1: Missing on-policy distillation baseline and comparison.
>
> We acknowledge that on-policy is indeed an effective distillation method, but it has significant limitations that affect its widespread application. Specifically, it **requires the tokenizers and vocabularies of the student model and the teacher model to be consistent** in order to calculate the loss. However, in many scenarios (including the experimental setup of this paper), the teacher model and the student model belong to **different families of LLMs**, making on-policy distillation inapplicable. In particular, the method proposed in this paper and the compared baselines are all data-based distillation methods, which can work **even if the teacher model is a closed-source model**. In contrast, the on-policy method cannot be used with closed-source teacher models.
>
> > Response to W2 & Q5: Analysis of student queries and teacher feedback
>
> - **Sub-questions vs. Final Answers:** As shown in **Figure 4**, the average number of queries per trajectory during the peak learning phase is greater than 1. This indicates the student is not merely asking for the final answer but is engaging in multi-step reasoning by asking sub-questions.
> - **Learning Mechanism:** Even if a student asks for a solution, they cannot simply "copy" the teacher's internal state because we strip the teacher's "thinking" tokens (e.g., content within `<think>` tags) before returning the result. The student receives only the conclusion (observation) and must internally deduce the thought required to reach that conclusion. This forces "active distillation" rather than passive imitation.
> - **Query Novelty:** We analyzed the student's queries (Line 437) using GPT-4o and found that the majority of generated queries are *not* equivalent to the original problem, confirming that the student is actively decomposing problems rather than repeating the prompt.
>
> | **Step** | **20** | **60** | **100** |
> | --- | --- | --- | --- |
> | Equivalent (%) | 4.37 | 3.12 | 1.18 |
>
> > Response to Q2: Explanation of the temperature coefficient
>
> The temperature coefficient mentioned refers to the sharpening of the teacher's distribution in our design of **Amending Importance Sampling Coefficient**. We treat the teacher's distribution as a one-hot distribution (**temperature** $\rightarrow 0$) to reduce computational complexity and resolve vocabulary inconsistencies.
>
> > Response to Q3: Experimenting with a weaker teacher and stronger student
>
> The scope of this paper is specifically **distillation,** transferring capabilities from a strong model to a weaker one. If the student is already stronger than the teacher, distillation is theoretically unnecessary. In such scenarios, self-play methods (like RLVR on the student itself) would likely be more effective than seeking guidance from a weaker teacher.
>
> > Response to Q4: Does the student also learn from the think tokens or only the response tokens?
>
> Although our teacher LLMs are thinking LLMs, when we return the responses of teacher LLMs to student LLMs, we only retain the conclusion part of the responses (e.g., removing the content within the `<think>` `</think>` tags). Therefore, student LLMs will only learn from the response tokens.
>
> ---
>
> Thank you for your insightful review. We hope this addresses your concerns and remain open to further discussion.

---

> > ### Comment · Reviewer_8BPw · 2025-11-27
> >
> > Thank you for your response! Two of my primary concerns still remain which are:
> >
> > * While I agree on the point that on-policy distillation cannot be applied on top of closed teacher LLMs, it still should not stop the authors from experimenting with a setting where that's not the case. This is necessary to understand how the proposed method compares to the most prevalent way of doing distillation (given recent studies like https://thinkingmachines.ai/blog/on-policy-distillation/).
> >
> > * Let me know if I've missed it but I did not see any qualitative examples of neither the student's questions nor the teacher's feedback. Given the interactive nature of the proposed method, reward hacking is always a possibility and without such analysis, it's hard to build trust on the method.
> >
> > In light of these concerns, I'll maintain my score.

---

> > > ### Author Response · Authors · 2025-11-29
> > >
> > > Thank you for your reply. Here are our supplementary responses.
> > >
> > > > Comparison with on-policy distillation
> > >
> > > Given that on-policy distillation paradigms are typically constrained to LLMs with same architectures, we conduct supplementary experiments using Qwen2.5-7B-Instruct as the student and Qwen2.5-32B-Instruct as the teacher. As shown in the table below, AgenticDistillation **outperforms the on-policy baseline**, and AgenticDistillation also demonstrates robust effectiveness in cross-architecture distillation scenarios as demonstrated in the paper.
> > >
> > > | |**Math (avg)**|**Science**|**Code**|**Puzzle**|
> > > | --- | --- | --- | --- | --- |
> > > | **Original** | 25.26 | 33.33 | 37.19 | 9.63 |
> > > | **SFT (off-Policy)** | 27.58 | 29.89 | 32.26 | 11.27 |
> > > | **On-Policy** | 28.37 | 34.53 | 39.18 | 16.28 |
> > > | **AgenticDistillation** | **29.62** | **36.12** | **42.17** | **20.44** |
> > >
> > > > Qualitative example
> > >
> > > We have added relevant qualitative examples in Appendix B.4 of the revised version, and these examples demonstrate that AgenticDistillation can indeed pose meaningful questions to learn from interactions.

---

### Author Response · Authors · 2025-11-19
**General Response**

We sincerely thank all reviewers for their thorough and insightful reviews. We are encouraged by the recognition of our work's novelty in agentic distillation and its potential as a future-proof framework. We also appreciate the constructive feedback regarding baselines, experimental setups, and efficiency, which has helped us significantly improve the quality and rigor of our paper. We have provided a detailed response to each reviewer. We have revised the paper according to the reviews, and the edits have been highlighted in **BLUE**. Here, we highlight our major clarifications regarding common concerns raised by multiple reviewers.

- **Applicability of Data-Based vs. On-Policy Distillation** (Reviewers 8BPw, 3uKu): We clarify that while on-policy distillation (typically minimizing KL divergence) is effective, it imposes strict constraints: it **requires the student and teacher to share identical tokenizers and vocabularies**. In our cross-family settings (e.g., Qwen vs. Llama) or scenarios involving **closed-source teachers**, logit-based methods are inapplicable. Our proposed data-based distillation serves as a universal bridge using natural language, enabling distillation across diverse architectures and from black-box models.
- **Test-Time Behavior and Generalization** (Reviewers fuv7, 3uKu, qKrD): A primary concern was whether the student relies on interaction during inference. We explicitly clarify that **interaction is solely a training scaffold**. At test time, the student model performs standard Chain-of-Thought (CoT) reasoning **without** accessing the teacher or generating feedback-style queries. We have validated this on zero-shot benchmarks (see **Appendix A.3**), demonstrating that the student has internalized the reasoning capability rather than overfitting to the interaction format.
- **Baselines and Experimental Rigor** (Reviewers 3uKu, qKrD): To address concerns about baseline strength, we have included an **"RL+SFT"** baseline (SFT on teacher trajectories followed by RL). Our method, Agentic Distillation, consistently outperforms this strong baseline (e.g., **14.82 vs 12.13** on AIME24). Regarding discrepancies with technical reports, we emphasize that we report the **average** performance over multiple runs to ensure statistical reliability, whereas technical reports often use `pass@1` or specific sampling hyperparameters. We ensured all models were evaluated under strictly identical settings for fairness.
- **Training Efficiency vs. Inference Efficiency** (Reviewer fuv7): While interactive training introduces computational overhead, we highlight that the primary goal is **inference efficiency**. The resulting student model is small, fast, and independent of the teacher during deployment. Furthermore, our method is more token-efficient than Rejection Sampling approaches because the student queries selectively rather than requiring the massive teacher model to generate full trajectories for every instance.

---

### Note · Authors · 2026-01-06

I have read and agree with the venue's withdrawal policy on behalf of myself and my co-authors.